# The Need for Biomarkers in the ALS–FTD Spectrum: A Clinical Point of View on the Role of Proteomics

**DOI:** 10.3390/proteomes11010001

**Published:** 2023-01-09

**Authors:** Francesca Vignaroli, Angelica Mele, Giacomo Tondo, Veronica De Giorgis, Marcello Manfredi, Cristoforo Comi, Letizia Mazzini, Fabiola De Marchi

**Affiliations:** 1Neurology Unit, Maggiore della Carità Hospital, 28100 Novara, Italy; 2Department of Neurology, S. Andrea Hospital, University of Piemonte Orientale, 13100 Vercelli, Italy; 3Department of Translational Medicine, University of Piemonte Orientale, 28100 Novara, Italy; 4Center for Translational Research and Autoimmune and Allergic Diseases (CAAD), University of Piemonte Orientale, 28100 Novara, Italy

**Keywords:** neurodegenerative diseases, amyotrophic lateral sclerosis, frontotemporal dementia, proteomics, biomarkers

## Abstract

Frontotemporal dementia (FTD) and amyotrophic lateral sclerosis (ALS) are severely debilitating and progressive neurodegenerative disorders. A distinctive pathological feature of several neurodegenerative diseases, including ALS and FTD, is the deposition of aberrant protein inclusions in neuronal cells, which leads to cellular dysfunction and neuronal damage and loss. Despite this, to date, the biological process behind developing these protein inclusions must be better clarified, making the development of disease-modifying treatment impossible until this is done. Proteomics is a powerful tool to characterize the expression, structure, functions, interactions, and modifications of proteins of tissue and biological fluid, including plasma, serum, and cerebrospinal fluid. This protein-profiling characterization aims to identify disease-specific protein alteration or specific pathology-based mechanisms which may be used as markers of these conditions. Our narrative review aims to highlight the need for biomarkers and the potential use of proteomics in clinical practice for ALS–FTD spectrum disorders, considering the emerging rationale in proteomics for new drug development. Certainly, new data will emerge in the near future in this regard and support clinicians in the development of personalized medicine.

## 1. Introduction

Frontotemporal dementia (FTD) and amyotrophic lateral sclerosis (ALS) are severely debilitating and progressive neurodegenerative disorders. FTD is the third-most common cause of dementia across all age groups and is the leading cause of early-onset dementia [1]. Clinically, FTD is a heterogeneous disorder characterized by various behavioral, language, and motor symptoms [2]. The clinical syndromes reflect the anatomical distribution of pathology, which involves the frontal and temporal lobes. FTD is thus considered part of the frontotemporal lobe degeneration spectrum of disorders, including also the atypical parkinsonism of progressive supranuclear palsy, corticobasal degeneration, and the complex FTD associated with motor neuron disease (FTD–MND) [3]. ALS is the most common motor neuron disease [4]; it is clinically characterized by motor dysfunction due to upper and lower motor neuron degeneration that leads to progressive paralysis and death from respiratory failure, usually within three-to-five years after the onset of symptoms [4,5]. Although motor dysfunction is the cardinal symptom of ALS, up to 50% of these patients developed cognitive impairment and, notably, roughly 15% of these patients presented clinical symptoms of FTD [6].

A distinctive pathological feature of neurodegenerative diseases, including ALS and FTD, is the deposition of aberrant protein inclusions in neuronal cells, which leads to cellular dysfunction and neuronal damage and loss [7]. To date, the biological process behind developing these protein inclusions is not well clarified. Furthermore, each neurodegenerative disease presents specific protein aggregation that accumulates in different cellular populations, damaging specific brain areas, and eventually leading to different clinical syndromes [8]. This evidence promoted a conceptual pathology-based approach in which neurodegenerative diseases, classified as proteinopathies, belong to a spectrum characterized by the deposition of specific pathological protein aggregates [9].

Since protein aggregation and inclusion formation are the pathological hallmarks of neurodegenerative diseases, understanding these processes is necessary to identify new disease mechanisms and potential treatment targets. In this context, proteomics methods are being applied to study neurodegeneration and identify useful disease biomarkers for diagnosis, prognosis, and therapy [10,11,12].

## 2. ALS–FTD Spectrum Disorders

### 2.1. Frontotemporal Dementia: Clinical and Pathogenesis

FTD includes neurodegenerative syndromes characterized by progressive behavioral, executive, and language deficits [13,14]. Based on the prevalent symptoms, FTD is classified into three clinical variants: the behavioral variant (bvFTD), and two language disorders classified as primary progressive aphasia (PPA), namely the non-fluent/agrammatic PPA (nfvPPA), and the semantic variant PPA (svPPA) [13]. 

bvFTD is the most frequent clinical syndrome, and it is characterized by early changes in behavior, personality, emotional modulation, and executive function [13]. Symptoms may include disinhibition, inappropriate touching, or over-familiarity with strangers [13]. There also be observed new onset of gambling, stealing, or making decisions without regard to the consequences [15,16,17,18]. Perseverated, stereotyped, or compulsive behaviors, often with ritualistic characteristics, can manifest in bvFTD patients, including simple or complex repetitive motor behaviors. In addition, speech may become stereotyped with repetitive patterns [13]. Further neuropsychiatric disturbances include apathy, lack of empathy, impaired social cognition, and dietary changes, such as a craving for sweets and hyperorality [13,19,20].

In PPA, deficit of language is the presenting symptom [21,22]. SvPPA is characterized by progressive loss of semantic knowledge, with the earliest symptoms represented by anomia and single-word comprehension deficits, whereas repetition, syntax, and grammar remain notably spared [22]. Neurodegenerative process can involve the temporal regions bilaterally or preferentially to either the left temporal lobe, which is associated with predominant semantic deficit, or the right, which is associated with behavioral disturbances; the different distribution of pathology alterations seems to be associated with different vulnerability patterns and large-scale network organization [23,24]. 

NfvPPA is firstly present with effortful speech production and word-finding problems. Then, the patient’s speech becomes slow and labored, with phonological errors, grammatical errors, and word retrieval difficulties [22].

From a pathological point of view, FTD belongs, together with the other frontotemporal lobar degeneration disorders, to the group of tauopathies. Tau, encoded in the microtubule-associated protein tau (MAPT) gene, is a neuron-specific protein that participates in binding and stabilizing the microtubules; in addition, it seems to have a role in axonal outgrowth and neuronal plasticity [25]. In FTD, the inclusions are formed of hyperphosphorylated tau. The phosphorylation of tau is needed for microtubule assembling and stability. However, when hyperphosphorylated, the balance between tau and microtubules is disrupted, and tau undergoes conformational changes and aggregation, thus leading to neuronal damage and death [26]. Abnormal tau deposits are the most common pathology in bvFTD and in nfvPPA [27]. In addition, mutations in the *MAPT* gene can be responsible for genetic form of FTD, which is often associated with parkinsonism [28,29]. 

Besides tauopathy, FTD may rely on the pathological deposition of ubiquitin-positive aggregates. The deposition of the TDP-43 is the neuropathological hallmark of most of the ubiquitin-positive, tau-negative FTD cases [30]. 

TDP-43 is an RNA-binding protein involved in RNA splicing, translation, mRNA transport, and miRNA processing. However, the exact role of TDP-43 in the brain is not yet known, but it is probably essential for neuronal development, axon guidance, and synaptic activity [26]. TDP-43 is mostly found in the nucleus of neuronal cells; its pathological form shows several modifications, such as hyperphosphorylation and ubiquitination, which lead to cytoplasmatic inclusions [31]. Four subtypes of TDP-43 have been described by their morphology and anatomical localization: TDP type A, B, C, and D [32]. BvFTD is associated with TDP-43 types A, B, and C; nfvPPA is frequently associated with the TDP-43 type A pathology; svPPA is frequently associated with TDP-43 type C pathology [32]. In addition, most FTD–MND cases are associated with TDP-43 pathology [33]. Finally, the ubiquitin-positive, tau-negative as well as TDP-43-negative FTD cases are associated with inclusions of the fused-in-sarcoma (FUS) protein [34,35]. 

FUS protein, similar to TDP-43, is a DNA/RNA-binding protein; its role is not fully understood, but it is likely involved in regulating DNA and RNA metabolism; it is crucial for neuronal structure and plasticity. FUS, like TDP-43, is generally localized in the nucleus, but in its abnormal form, it aggregates in the cytoplasm [36] and is associated with both FTD and ALS [34,37]. The different pathological phenotypes are summarized in Figure 1. 

### 2.2. Amyotrophic Lateral Sclerosis: Clinical and Pathogenesis

ALS is a rare and rapidly progressive neurodegenerative disease characterized by the degeneration of the upper and lower motor neurons, which leads to progressive voluntary muscular weakness. The site of symptoms’ onset varies between patients, and the disease can present as multiple phenotypes (i.e., spinal and bulbar). The disease rapidly results in impaired mobility, difficulty in daily activities, and need of constant assistance [38]. Patients may develop respiratory failure as the disease progresses, with a median survival time of 3–5 years from the onset of symptoms [4]. Notably, unlike what was believed in the past, ALS is not only a “motor disease”, but it is also frequently associated with extramotor symptoms, mainly behavioral and cognitive alterations. Cognitive impairment can occur early during the disease course, and it generally concerns 40–50% of people with ALS [6]. Cognitive deficits generally involve executive functions, attention, working memory, and organization dysfunction. Behavioral changes often manifest as personality changes, obsessions, and disinhibitions [39,40]. In 10–15% of patients, the cognitive impairment overlaps the above-described bvFTD, thus supporting the idea that FTD and ALS are part of the same clinical and neuropathological spectrum (ALS–FTD) [6]. 

Although the pathological mechanism behind ALS is largely unknown, the neuropathological hallmark of ALS is the development of detrimental cytoplasmic protein inclusions both in motor neurons and surrounding oligodendrocytes [41]. Most ALS patients (97%) present TDP-43 inclusions, especially those associated with a TDP-43 type B pathology [33,42]. As shown, TDP-43 proteinopathies are also linked to FTD in at least 50% of cases. The similarities in their neuropathological processes, which are associated with overlapped behavioral and cognitive tracts in ALS and FTD, leads to the hypothesis that these two neurodegenerative diseases are different manifestations of TDP-43 pathology [43]. In ALS, as in FTD, the FUS protein works similar to TDP-43 by moving from the nucleus and aggregating in the cytoplasm of ALS patients [41]. Other significant protein inclusions in ALS can be formed by misfolded SOD-1, an antioxidant enzyme that protects cells from the harmful effects of superoxide radicals [41,44]. However, compared to what was said for TDP-43 and FUS, how these protein inclusions result in motor-neuron degeneration remains unclear; it is hypothesized that the high-molecular-weight complexes that precede the formation of protein inclusions are the real toxic species rather than the final protein aggregates [45].

## 3. Biomarkers in ALS–FTD Spectrum Disorders

Biomarkers are objective parameters able to show biological alterations related to diagnosis and disease progression [46]. Several biomarkers were proposed in recent years, including genetic-, biochemical-, and imaging-based markers. To date, the diagnosis of ALS–FTD spectrum disorders is based on clinical symptoms supported by instrumental and laboratory-specific alterations. Cerebrospinal fluid (CSF) biomarkers hold a fundamental exclusionary role in differentiating neurodegenerative dementia subtypes [47]. Similarly, the genetic identification of specific pathogenic mutations allows the identification of familial forms of diseases, including carriers and presymptomatic individuals. A combination of genetic and imaging alterations could have clinical and prognostic effects [48]. The identification of reliable biomarkers supports the development of the movement toward precision medicine, aiding the diagnostic workup at the earliest stages of neurodegeneration in several neurodegenerative diseases together with a trustworthy prognostic prediction [49]. The need to integrate disease-specific biomarkers in clinical practice is strictly related to the possibility of developing effective disease-modifying therapies, helping the design of clinical trials, stratifying the patient population, individualizing therapeutic interventions, and predicting adverse drug reactions and the positive effects of drug treatments [50]. The ideal biomarker should be measurable with no variability, adaptable in response to therapy alterations, and presenting a large signal-to-noise ratio. Moreover, the value of a biomarker depends on the pathophysiological relationship between the biomarker and a specific clinical endpoint [50,51,52]. Adhibition of biomarkers in clinical practice and drug development for neurodegenerative diseases could help to demonstrate drug response and target engagement in these conditions [53]. Regarding ALS, no successful therapy can change the course of this pathology [54]. This is partly caused by the absence of specific biomarkers that could help identify individuals at risk of developing the disease [55]. Focusing attention on fluid biomarkers, neurofilament proteins [56,57] are between the most studied biomarkers on blood and CSF; in addition, interleukins and cytokines linked to microglia activation and inflammation [58,59] are of growing interest. It was found that levels of neurofilament in CSF and blood correlate with disease progression and survival in ALS patients [58,60]. Similarly, high levels of blood MCP-1, TNF-alfa, and GM-CSF were related to disease duration [61,62]. Furthermore, higher serum uric acid levels correlated with more prolonged survival in male patients and with better ALSFRS-R scores [63,64]. On the contrary, serum creatinine levels inversely correlate with ALSFRS-R scores and forced vital capacity [65,66]. The biomarkers mentioned above are just some of those proposed; it is important to underline that their clinical application depends on disease specificity [53]. Biomarkers could be a fundamental aid in stratifying the patient population of ALS and neurodegenerative diseases to identify subpopulations of patients in these heterogenous disorders, a concept introduced as precision medicine.

## 4. Proteomics and its Complexity

Multiomic approaches are being developed to guide decisions regarding disease prevention, diagnosis, and therapy. The primary aim of these techniques is to assess a complete characterization of individual disease risk [67]. They are represented by several techniques, which are summarized in Table 1. 

Mainly, proteomics tools allow characterization of the expression, structure, functions, interactions, and modifications of proteins of tissue and biological fluids including plasma, serum, and CSF [68,69]. The protein-profiling characterization aims to identify disease-specific protein alterations or specific pathology-based mechanisms which may be used as markers of neurodegenerative disease [70]. A schematic classification of proteomics tools reckons on conventional techniques, advanced techniques, quantitative techniques, and high throughput techniques followed by statistical and bioinformatics analysis [71]. Conventional techniques, including chromatography-based techniques, are usually used for protein purification, such as ion exchange chromatography, size exclusion chromatography, and affinity chromatography [71]. On the other hand, the quantitative analysis of a specific protein can be executed by an enzyme-linked immunosorbent assay (ELISA), which is able to quantify soluble substances in fluids, and Western blotting [72,73]. Western blotting is a procedure able to investigate, under proper control, the presence and relative abundance of post-translational modifications as well as to study protein–protein interactions [74,75]. 

Advanced proteomic techniques include protein microarray, liquid chromatography coupled with mass spectrometry (LC-MS/MS), and gel-based approaches. The latter category comprises techniques allowing the separation of complex protein samples, such as the sodium dodecyl sulfate-polyacrylamide gel electrophoresis (SDS-PAGE), two-dimensional gel electrophoresis (2-DE), and two-dimensional differential gel electrophoresis (2D-DIGE). LC-MS/MS guarantees higher sensitivity in analyzing complex protein mixtures [76]. Protein microarrays have been used for rapid expression analysis even though exploring the function of a complete genome remains challenging [71]. Lastly, Edman sequencing could be used to define the aminoacidic sequence of a specific protein [77].

Quantitative techniques include isotope-coded affinity tag (ICAT) labeling, stable isotope labeling with amino acids in cell culture (SILAC), and isobaric tags for relative and absolute quantitation (iTRAQ), which rely on chemical labeling reagents used for the quantification of proteins [78,79]. Another widely used method is label-free quantification: this technique determines relative quantities of proteins without using any labelling reagents. The three-dimensional structures of proteins, which can be correlated with their biological functions, can be obtained with techniques such as X-ray crystallography and nuclear magnetic resonance (NMR) spectroscopy. The data obtained from the above-mentioned methods are then elaborated using bioinformatic analysis; in recent years, standardized tools to analyze and study proteomic data have been developed. These methods exploit the pre-existing amount of biological knowledge in order to associate proteomic data to molecular processes and functions and, thus, generate new biological hypotheses [80].

We must remember that in the beginning, proteomics techniques were generated to detect a long list of different proteins [81]. Today, the primary aim is to generate hypotheses regarding new biological mechanisms or biomarkers and to make functional interpretations of the results. Moreover, the proteome is very versatile and changes depending on the fluid or tissue where it is analyzed. It also changes based on age, disease onset, and how it is measured. The other advantage is the selectivity and reliability of results [82]. 

## 5. Proteomics in the ALS–FTD Spectrum Disorders

Protein aggregate analysis, the identification of aggregated abnormal protein interactions and of proteins with an anomalous quaternary structure, may help to identify the pathological mechanisms involved in ALS–FTD. In recent years, several proteomic-based studies have been carried out on ALS–FTD spectrum disorders to explore the relevance of disease-related proteins and their potential roles in clinical practice. In ALS, aberrant protein folding and the formation of toxic protein aggregates are two crucial biological features. Indeed, the incorrect assembly of the protein in its native form leads to toxic molecules that potentially cause an overload of the degradation machine [83]. Moreover, the correct folding of proteins is crucial for their assembly into protein complexes and for their molecular interactions, which are both key elements of most of their physiological processes [84]. 

### 5.1. Proteomics in Cellular and Animal Models

The recent acceleration in the discovery of new genes related to ALS–FTD spectrum disorders led to the need to develop strategies to identify the molecular pathways and proteostasis dysfunctions related to them, taking advantage of cellular and animal models [85]. Regarding cellular models, different studies included mutant C9orf72, SOD1, and TDP-43 and aimed to reproduce some conditions which characterize ALS and FTD, such as protein aggregation, mitochondrial dysfunction, and cellular toxicity. 

Hartmann et al. [86] explored the neural interactions between the cytoplasmatic and nucleolar compartments in patients with C9orf72 mutations. They found that the overexpression of nucleolar aggregates associated with the C9Orf72 mutations reduced the number of synaptic proteins detected with proteomics. Other mechanisms associated with C9orf72 pathologies were discovered, such as the defects in stress granule homeostasis [87]. Boeynaems et al. [87] identified an active role for arginine-rich domains in this process, as they are able to induce a change in RNA and granule metabolism as well as spontaneous stress granule assembly. In addition, proteomics analysis of fibroblasts in ALS patients carrying the C9orf72 mutation revealed alterations in glucose metabolism and protein homeostasis. In fact, many proteins involved in the translation mechanism were powerfully downregulated in these cells compared with fibroblasts from wild-type ALS patients [88]. Motoneurons from C9orf72 patient-derived iPSCs have also altered mitochondrial axonal transport, impaired mitochondrial metabolism, and shorter axons [89]. 

Similarly, other cellular studies identified SOD1 and TDP-43 [90,91] protein interactions. McAlary, using a cell-based assays approach, showed that the aggregation of SOD1 variants is well-correlated to cellular toxicity even without a subsequent correlation with disease severity. Instead, in regard to TDP-43, the group of Rogelj [91] indicated that TDP-43 is an important regulator of RNA metabolism and intracellular transport in ALS–FTD, observing that proteins related to cellular processes (Ran-binding protein 1, DNA methyltransferase 3 alpha and chromogranin B) were downregulated upon TDP-43 knockdown.

Animal models can also validate these genetic and biological alterations [85]. In astrocytes from the ALS mouse model overexpressing human SOD1(G93A), a correlation between proteome and secreted metabolome involved in glutathione metabolism was observed. This finding has been speculated to be responsible for altered astrocyte functions due to a depletion of proteins and secreted metabolites [92]. 

In addition, in SOD1(G93A) in murine spinal cords, the interactor of misfolded SOD1 (e.g., HSPA8 and Na+/K+ATPAse-alfa3) had impaired activity that contributed to motor neuron vulnerability [93]. Furthermore, considering the mutated zebrafish, mutations in cyclin F were observed, which also provided high levels of activated caspase-3 and other proteins negatively involved with cellular survival [94]. However, due to the absence of precise animal models carrying the other mutations mentioned, there are no significant proteomics studies in other animal models. 

### 5.2. Proteomics in Human Samples

#### 5.2.1. Cerebrospinal Fluid

CSF represents a potential source of biomarkers because it is in contact with the brain’s interstitial fluid. In addition, changes in the CSF protein content can reflect alterations in proteins’ expressions within the central nervous system [95]. Various studies explored CSF’s potential diagnostic biomarkers for the ALS–FTD spectrum using proteomic analysis and by comparing data in patients and controls to characterize the proteomic profile of ALS–FTD samples. 

The pioneering study of Ranganathan and colleagues [96] compared the proteomic CSF profile of ALS patients and controls using surface-enhanced laser desorption ionization-time of flight mass spectrometry. Three major CSF biomarkers were identified that were significantly different between patients and controls: the carboxy-terminal fragment of neuroendocrine protein 7B2, which was increased in patients and is involved in the maturation and release of hormones, neuropeptides, and growth factors; transthyretin and cystatin C, which were decreased in patients and are involved in neuroprotection and extracellular proteins homeostasis. Several subsequent studies analyzing the CSF proteome profile with different techniques revealed panels of candidate biomarkers in ALS, including zinc- and iron-binding proteins involved in many metabolic processes [97,98] as well as proteins associated with synaptic regulation, apoptosis, extracellular matrix regulation, and neuroinflammation [99,100,101]. 

The analysis of proteome profiles has been tested in the diagnostic workup of ALS compared with controls and patients affected by other neurological diseases [102,103,104,105,106], showing differentiation between the profiles of ALS patients and non-ALS conditions with good sensitivity and excellent specificity [102,107]. 

An early study in FTD patients compared with controls identified significant differences in several CSF proteins, including the Zn-alpha-2-glycoprotein, whose levels were increased in patients [108]. Intriguingly, the same findings were reported a few years later in ALS patients in the study of Brettschneider and colleagues, again suggesting possible common pathogenic protein profiles along the ALS–FTD spectrum [98]. In FTD, the proteomic approach has been shown to potentially aid the differential diagnosis. A comparative proteomic analysis documented differences in the expression of CSF protein profiles in FTD patients compared to controls and Alzheimer’s disease patients, thus suggesting a different pathophysiological background between the two dementia disorders [109]. As for ALS, profiling CSF proteins in genetic FTD cases may aid in tracking pathophysiological changes during different disease phases. A proteomic approach using mass spectrometry was used in presymptomatic and symptomatic carriers of the granulin (GRN) mutation and healthy noncarriers. Differences in CSF protein expression were not only revealed between symptomatic GNR-mutated and noncarriers, but also between presymptomatic and symptomatic GNR-mutated carriers, including proteins involved in synaptic activity, vesicle secretion, and inflammatory responses [110].

A key point for proteomic studies is the possibility of detecting biomarkers for disease progression and prognostic value. A longitudinal study analyzing the CSF of 14 ALS patients using data-independent acquisition mass spectrometry and evaluating data through mathematical modeling identified changes in 28 peptides involved in stress response and innate immunity as fixed effects in disease progression [111]. Several proteins’ CSF changes have been associated with disease severity, disease progression, and survival [95,96,105]), but the importance of their prognostic role still requires further confirmation. The studies on cerebrospinal fluid are summarized in Table 2. 

#### 5.2.2. Blood

Few studies applied proteomic analysis to identify changes in serum proteins in the ALS–FTD spectrum and mainly focused on exploring neuroinflammatory responses and their corresponding peripheral mechanisms. Neuroinflammation is widely recognized as a common mechanism in neurodegenerative conditions, representing a promising target for modulatory therapies aiming to stop or slow neuronal loss [112]. Cao and colleagues [113] compared more than one hundred markers of inflammation, including cytokines, growth factors, and blood–brain barrier breakdown markers in the serums of ALS patients to controls. The authors identified the 20 most changed proteins, which were mainly represented by proangiogenic and growth factors, thus suggesting that altered glial activation and blood–brain barrier leakage may be involved in ALS pathogenesis. The detection of differences in the serum levels of acute phase reactants in ALS patients than in controls has been reported by another study [114], together with changes in lipid homeostasis proteins, thus supporting the hypothesis of a metabolic shift towards increased peripheral use of lipids in ALS patients and suggesting the involvement of lipid homeostasis in the disease [114]. Serum protein changes were also correlated with specific characteristics of the disease. ALS patients with cognitive impairment showed a different serum proteomic profile than ALS patients without cognitive impairment, especially involving proteins within the coagulation and immune pathways, confirming the utility of proteomic analysis as a tool to study disease-specific features [115]. Lastly, protein changes in the serum of both FTD and ALS patients compared to controls were analyzed in the study of Katzeff and colleagues [116]. The authors found 23 serum proteins, mainly involved in innate immunity and calcium signaling, dysregulated in bvFTD patients and 14 in ALS patients as compared to controls. Intriguingly, 11 of these proteins were altered in both diseases, suggesting possible common pathophysiology pathways between ALS and FTD [116]. The studies on blood are summarized in Table 2. 

**Table 2 proteomes-11-00001-t002:** Cerebrospinal fluid and blood studies on ALS–FTD spectrum disorders: methodology and main findings.

Disease	Year	Method	Main Findings
Cerebrospinal Fluid			
ALS vs. HC	2005	surface-enhanced laser desorption ionization time-of-flight mass spectrometry	- protein 7B2 increased in patients - transthyretin and cystatin C decreased in patients [96]
ALS vs. HC	2012	two-dimensional difference in gel electrophoresis with matrix-assisted laser desorption ionization time-of-flight mass spectrometry	- parkin-like and iron and zinc binding proteins increased in patients [97]
ALS (fast vs. slow)	2010	two-dimensional difference in gel electrophoresis with matrix-assisted laser desorption ionization time-of-flight mass spectrometry	- heat shock protein 1, alpha-1 antitrypsin, fetuin-A precursor, transferrin, transthyretin, and nebulin-related anchoring protein were higher in fast progressors [98]
ALS vs. HC	2022	ultra-sensitive proximity extension assay	- junctional adhesion molecule A protein, tumor necrosis factor receptor 2, and chitinase 1 were upregulated in patients - myoglobin was downregulated in patients [99]
ALS vs. HC	2013	liquid chromatography-tandem mass spectrometry	- elevated levels of chitotriosidase in patients [100]
ALS vs. HC	2012	paramagnetic bead chromatography with matrix-assisted laser desorption ionization time-of-flight mass spectrometry	- upregulation of secreted phosphoprotein 1 in patients [101]
ALS vs. HC and other neurodegenerative diseases	2015	label-free liquid chromatography-tandem mass spectrometry	- pathways altered for protein 63, amyloid-like protein 1, SPARC-like protein 1, and cell adhesion molecule 3 in ALS patients [102]
ALS vs. other neurological diseases	2020	liquid chromatography-tandem mass spectrometry	- CXC motif chemokine ligand 12 increased in patients [103]
ALS vs. HC and Parkinson’s disease	2019	targeted multiple reaction monitoring (MRM) mass spectrometry	- levels of ubiquitin carboxy-terminal hydrolases, such as protein 1, glycoprotein non-metastatic melanoma protein B, and cathepsin D were increased in patients [104]
ALS vs. HC and other neurodegenerative diseases	2016	two-dimensional liquid chromatography mass spectrometry	- insulin-like growth factor II was significantly downregulated in ALS patients - glutamate receptor 4 was significantly upregulated in patients [105]
ALS vs. HC and neuropathies	2008	two-dimensional gel electrophoresis	- differential expression of ceruloplasmin isoforms in ALS patients compared to HC - increase in the relative abundance of more basic ceruloplasmin forms, corresponding to nonsialylated proteins in patients [106]
ALS vs. other neurological diseases	2009	Bio-Plex human 27-plex panel of cytokines and growth factors with atomic absorption spectroscopy	- a panel of interleukins (i.e., IL6, IL2, IL16, and IL17) were higher in ALS compared to others [107]
ALS	2020	shotgun proteomics and data-independent acquisition mass spectrometry	- in a longitudinal follow up, changes in abundance from 28 peptides [111]
FTD vs. HC	2004	prefractionation method with two-dimensional electrophoresis	- Zn-alpha-2-glycoprotein increased in patients [108]
FTD vs. HC and AD	2002	Two-dimensional gel electrophoresis with mass spectrometry	- granin-like neuroendocrine precursor, pigment-epithelium derived factor, retinol-binding protein, apoE, haptoglobin, and albumin levels altered in FTD patients [109]
FTD (GRN carriers vs. non-carriers)	2019	parallel reaction monitoring mass spectrometry	- symptomatic GRN mutation carriers had lower levels of neuronal pentraxin receptor, receptor-type tyrosine-protein phosphatase N2, neurosecretory protein VGF, chromogranin-A, and V-set and transmembrane domain-containing protein 2B than presymptomatic carriers and noncarriers [110]
Blood			
ALS vs. HC	2022	cytometric bead array and proteome profiling	- fractalkine, BDNF, EGF, PDGF, Dkk-1, MIF and angiopoietin-2, S100β were unchanged in ALS serum [113]
ALS vs. HC	2017	bi-dimensional electrophoresis and mass spectrometry	- acute phase reactants and lipid homeostasis proteins were higher in ALS [114]
ALS vs. HC	2018	nano-liquid chromatography and time-of-flight mass spectrometry	- the LXR/RXR and coagulation pathways were downregulated in LAS - the complement pathway was upregulated - differences between ALS patients with and without cognitive impairment [115]
ALS vs. FTD vs. HC	2020	nano-capillary liquid chromatography–tandem mass spectrometry	- 23 proteins were altered in FTD vs. HC (increased: APOL1, C3, CTSH, EIF5A, MYH2, S100A8, SUSD5, WDR1; decreased: C1S, C7, CILP2, COMP, CRTAC1, EFEMP1, FBLN1, GSN, HSPG2, IGHV1, ITIH2, PROS1, SHBG, UMOD, VASN) - 14 proteins were altered in ALS vs. HC (increased: APOL1, CKM, CTSH, IGHG1, IGKC, MYH2; decreased: C7, COMP, CRTAC1, EFEMP1, FBLN1, GSN, HSPG2, SHBG) [116]

CSF: cerebrospinal fluid; ALS: amyotrophic lateral sclerosis; HC: healthy controls; FTD: frontotemporal dementia; AD: Alzheimer’s disease; GRN: progranulin.

#### 5.2.3. Other Tissues

As previously said, ALS and FTD are pathologically characterized by the presence of protein inclusions due to dysregulation in protein expression, processing, or degradation. In light of this, the proteomic analysis of post-mortem samples, such as from the cortex and spinal cord, can help to delineate the molecular changes in protein composition in ALS–FTD patients. 

In 2011, Gozal et al. conducted a proteomic analysis of hippocampal dentate granule cells in sporadic FTD subjects by using a combined approach consisting of laser capture microdissection and high-resolution liquid chromatography-tandem mass spectrometry. Compared to controls, they identified 1252 proteins in hippocampal dentate granule cells of FTD patients. Additionally, SEPT11, a protein associated with the cytoskeleton, was a component of protein inclusions with the well-known TDP-43 protein. These findings highlighted the cytoskeleton-associated protein’s possible role in FTD pathogenesis [117,118]. Interestingly, the analysis also showed that proteins not associated with protein inclusions presented a dysregulated expression [117,118]. 

Instead, to better understand the pathogenic role of the reduction of C9orf72 expression in FTD patients, a proteomic approach was applied to determine the level of reduction in the long and short isoforms of C9orf72 in the frontal cortices of mutated patients. The results showed that the C9Orf72 long isoform was significantly decreased in the frontal cortices of genetic patients compared to normal subjects [119].

In ALS patients, spinal cord protein profiles revealed dysregulated expression in proteins involved in mitochondrial, calcium, and protein metabolism. In particular, ATP5D (a subunit of ATP synthase that is essential for ATP production) was reduced mainly at synapses, supporting the role of synaptic dysfunction in ALS pathogenesis. In addition, the level of calmodulin, a protein implicated in calcium metabolism, was downregulated, which determined the disruption of calcium homeostasis [120]. Moreover, protein acetylation seems to be differentially regulated: for example, the glial fibrillary acid protein (a GFAP-component of the filament of astrocytes that plays a role in astrocyte-neuron interaction) was found to be heavily acetylated and upregulated in an ALS patient’s spinal cord, suggesting a potential neuroprotective effect of histone deacetylase inhibitors [121]. 

The proteomic analysis of post-mortem samples was also applied to investigate the molecular basis of the pathological overlap between ALS and FTD. With an elegant study published in 2018, Umoh et al. showed the comparison of protein expression in frontal cortical tissue from post-mortem ALS, FTD, and ALS–FTD cases, revealing different coexpressed proteins involved in synaptic transmission, inflammation, and RNA metabolism across the ALS–FTD spectrum. Furthermore, ALS cases carrying the C9orf72 mutation presented an increase in proteins associated with astrocytes and microglia compared to sporadic cases, implying that genetic expansion could also alter the inflammatory response [43]. In addition, Iridoy et al. in the same year compared protein composition in the spinal cords of ALS and FTD patients, showing that ALS and FTD partially shared molecular and functional alterations with a common impairment in mitochondrial metabolism. However, parts of the altered protein expression, such as galectin 2, transthyretin, and protein S100-A6 for ALS, remained disease-specific [122]. 

## 6. Discussion

Despite the increasing efforts in proteomics research and the undeniable need for reliable biomarkers in the neurodegenerative field, consolidated fluid biomarkers for the ALS–FTD spectrum must still be improved, often due to preliminary or contrasting results. Additionally, no treatment monitoring the use of biomarkers is validated, which limits the development of precision medicine. This aim seems far-reaching, but it is undoubtedly a point on which much translational research focuses. 

As described above, the pathogenic mechanisms underlying neurodegeneration are challenging to understand for several reasons. First, the complexity of the human genome and the significant number of gene-related proteins involved in the pathological disease mechanisms hinder the identification of a single pathway on which we must act. In addition, protein molecules differ for several variables, such as the patients’ origins, innate and acquired genetic variability (e.g., related to post-translational modifications), and the different cellular tissues primarily affected. Combining these factors generates distinct protein patterns able to modulate different biological processes. Although two-dimensional gel electrophoresis first discovered the complexity of these protein patterns, newer proteomic techniques are adding essential details in this regard. However, these proteins can occur with different isoforms (recently called proteoforms) within the same disease and between different neurodegenerative disorders. One example is symbolic in this regard: the protein tau. Six isoforms of tau protein are expressed in the human brain. They derive from alternative splicing of the MAPT gene. The alternative splicing involves exons 2, 3, and 10, and it generates isoforms differing in the number of repetitions of the C-terminal microtubule-binding domain. The exclusion of exon 10 produces isoforms with three repetitions (3R), whereas the inclusion of exon 10 produces isoforms with four repetitions (4R) [123]. Tau isoforms also differ regarding the presence of 0, 1, or 2 N-terminal inserts. In addition, numerous post-translational modifications, including phosphorylation, acetylation, methylation, and glycosylation, may induce further variability in the expression and function of tau isoforms [124]. Phosphorylation inhibits tau binding to microtubules, interfering with their stabilization and axonal transport. Hyperphosphorylation may also induce misfolding of the protein with consequent production of abnormal aggregates. These aggregates initially polymerize into protofibril filaments with a β-sheet structure in the form of straight filaments, paired helical filaments, or twisted filaments. Misfolded tau gains a toxic function that triggers neuronal death [125]. Different tau isoforms, characterized by a specific molecular structure, precipitate in different tauopathies, and each tauopathy may display a specific distribution pattern in different brain areas.

Interestingly, the ratio of 4R to 3R tau isoforms is approximately one-to-one in the normal adult brain, but this ratio loses balance in neurodegenerative tauopathies, including FTD. Lastly, one must bear in mind that similar pathologies may determine different clinical phenotypes, and the same clinical manifestation can be due to different pathological aggregates. As an example, both 3R and 4R tau isoforms are present in FTD and Alzheimer’s disease dementia in straight filaments and paired helical filaments, but involving different vulnerable brain regions and reflecting different phenotypes [126,127]. 

Recent advancements in proteomic techniques could help in this search, and several advantages of proteomics make it a good choice for biomarker research. First, proteomics is a maturing discipline able to identify proteins relevant to neurodegenerative disorders. In fact, despite significant advantages in the genetic field, roughly half of the protein-coding genes lack a clear role at the protein level. In addition, several post-translational protein changes can alter the intrinsic function of specific disease mechanisms. Additionally, an interest in proteomic analysis may also be to evaluate the differential expression of specific proteins in different conditions, such as in different diseases or disease stages, to evaluate the roles of external or internal stimuli in alterations of intracellular signaling pathways. The advantages of proteomics as well as a comparison with genome and transcriptome techniques are summarized in Figure 2. 

Several critical issues in proteomic studies must still be addressed. One of the major criticisms is the validation of the identified proteins across parallel studies, which is essential to obtain conclusive evidence. In addition, the use of several techniques and the application of methodologies in several laboratories, even using the same tools, revealed a non-negligible biological heterogeneity, which hampers the validation of these methods for clinical purposes. 

## 7. Conclusions

Our narrative review aimed to highlight the need for biomarkers and the potential use of proteomics in clinical practice, considering the emerging rationale in proteomics for new drug development. Certainly, new data will emerge in the near future in this regard to supportclinicians in the development of personalized medicine.

## Figures and Tables

**Figure 1 proteomes-11-00001-f001:**
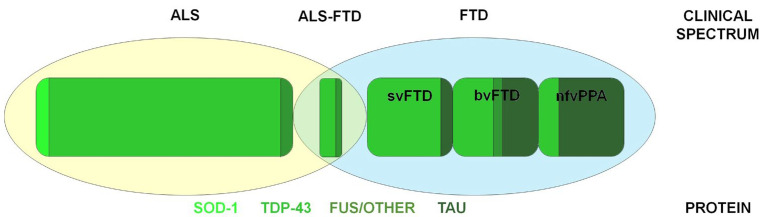
Clinical and pathological characteristics in the ALS–FTD spectrum. The figure represents the relative rate of protein deposits in the clinical subtypes. Most ALS patients present TDP-43 inclusions. SOD-1 alterations are typical in genetic ALS. TDP-43 is also associated with most svPPA cases and about 50% of bvFTD cases. The other most represented neuropathology substrate in bvFTD is represented by tau deposition, which also characterizes nfvPPA’s forms. ALS, ALS–FTD, and bvFTD are infrequently associated with FUS and other pathologies. Abbreviations: ALS, amyotrophic lateral sclerosis; bvFTD, behavioral variant FTD; FTD, frontotemporal dementia; FUS, RNA-binding protein FUS; nfvPPA, nonfluent variant primary progressive aphasia; SOD-1, superoxide dismutase type 1; svPPA, semantic variant primary progressive aphasia; TDP-43, transactive response DNA-binding protein 43.

**Figure 2 proteomes-11-00001-f002:**
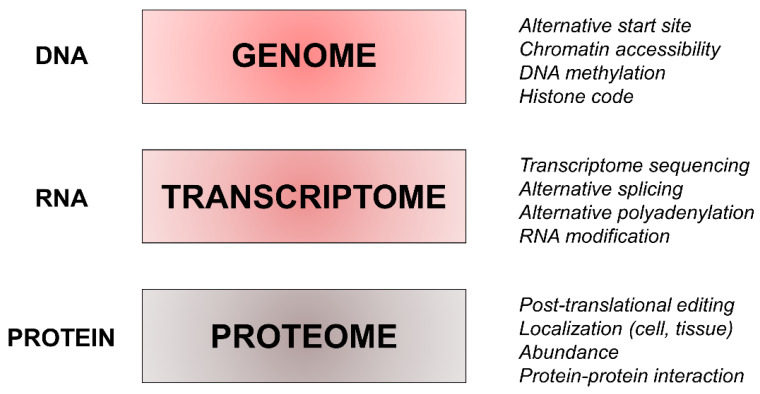
The main factors explaining the variability in gene expression, RNA synthesis, transcription, and protein expression and function. DNA can express variable coding potential using alternative transcription start sites, modulating DNA methylation, and chromatin accessibility. Alternative splicing can occur at the transcriptional level, and post-transcriptional editing of RNA increases variability in protein expression. Lastly, localization, abundance, and post-translational modifications influence protein function, which can also be affected by how proteins interact.

**Table 1 proteomes-11-00001-t001:** Multiomic approaches and their primary function.

Technique	Primary Function
**Proteomics**	characterization of protein constituents in biological samples - clinical proteomics: analysis of the proteins’ role in disease onset and progression - structural proteomics: evaluation of the protein structure related to its physiological and pathological role - functional proteomics: study of protein interactions
**Genomics**	set of DNA sequences provided by genome-wide association studies and, more recently, next-generation whole exome and whole genome sequencing data
**Transcriptomics**	representing gene expression patterns
**Metabolomics**	characterization of metabolic profiles
**Lipidomics**	characterization of the complete collection of lipids
**Epigenomics**	profile of the modifications to DNA that control gene expression
**Exposomics**	the sum of exposure an individual incurs over a period of time
**Microbiomics**	characterization of the microbes that reside in or on an individual

## Data Availability

Not applicable.

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
