# Peer review of "The Need for Biomarkers in the ALS–FTD Spectrum: A Clinical Point of View on the Role of Proteomics"

_proteomes, 2023, doi:10.3390/proteomes11010001_

Round 1

Reviewer 1 Report

The review entitled (The need for biomarkers in the ALS-FTD spectrum: a clinical point of view on the role of proteomics) submitted by Vignaroli et al., discusses both frontotemporal dementia (FTD) and amyotrophic lateral sclerosis (ALS) as forms of neurodegenerative diseases that gradually worsen over time and can be quite disabling. The accumulation of abnormal protein inclusions in neuronal cells is a distinguishing pathogenic hallmark of various neurodegenerative illnesses, including amyotrophic lateral sclerosis (ALS) and frontotemporal dementia (FTD). This results in cellular malfunction as well as neuronal damage and death. Despite this, to this day, the biological process responsible for creating these protein inclusions needs to be better understood, making the development of a treatment that modifies the condition impossible. Proteomics is a sophisticated method that may be used to analyze the expression, structure, functions, interactions, and changes of proteins found in tissue and biological fluids. These fluids include plasma, serum, and cerebrospinal fluid, among others. The purpose of the protein profiling characterization is to locate disease-specific protein alterations or particular pathology-based mechanisms that have the potential to serve as markers for these disorders. Considering the developing justification in proteomics for new medication development, The review highlights the need for biomarkers and the possible application of proteomics in clinical practice. I would like o to thank the authors for this comprehensive review, but a summative figure should be included at the end of the discussion in order to be accepted for publication.

Author Response

We thank the reviewer for the comment, which helps to improve the review quality. We followed the comment and added two comprehensive figures in the review (Figure 1 and 2). The first one is about the role of proteins in the ALS-FTD spectrum, the second one is a comparison between the main omics. 

Reviewer 2 Report

This manuscript reviews the potential of proteomic approaches to identify biomarkers for amyotrophic lateral sclerosis and frontotemporal dementia. After an introduction to ALS and FTD spectrum disorders, proteomic approaches/technology  to identify biomarkers are described. This is followed by reviewing studies involving proteomics for the characterization and identification of differences between ALS/FTD patients or model systems and controls. Finally, studies analyzing CSF, blood, and further human tissues are reviewed. Unfortunately, validated biomarkers for ALS or FTD have not been identified leaving this review with a disappointing hope for further improvements.

In general, the review summarizes the current state very well. However, I would have wished for a table or similar overview describing which potential biomarkers were proposed by individual studies and their limits.

ELISA was developed as a quantitative test and is routinely used to determine quantities of various  substances (hormones, proteins). Similarly, Western blot analysis permits - under proper control- quantitative analysis. The respective description should be modified (l226-228).

The language should be carefully checked to eliminate minor grammatical and typographical errors.(l26: Certainly; 134 omit the; 146 of the replace with that149 compared; 153 omit Approximately; 173 neurodegenerative; 177 omit the; 225 of a specific protein; 276 nucleoloar compartments; 280 induce; 285 expression wild-type ALS patients; and further to be corrected).

Author Response

We thank the reviewer for the dedicated time. As suggested, we added the following: 

1) two summarizing figures (in the manuscript as Figure 1 and 2), 

2) a table with the main findings for each study (in the manuscript as Table 2). 

Also, we corrected the paragraph regarding ELISA and Western blot, and we tried to revise all typos and similar mistakes. 

Reviewer 3 Report

This is an interesting review of current evidence from proteomic studies assessing the ALS-FTD spectrum of disorders. Although the topic is of interest and holds significant relevance for the understanding and definition of biomarkers in this context, there are some shortcomings in its organization that limit an easy comprehension of the main novelties here described.

-First of all, introductory paragraphs 2 and 3 are too long and general, out of focus as regards the declared topic of the review. These should be shortened.

-I would suggest the use of summarizing paragraphs and/or tables to help the reader  recapitulating the findings of all studies in an organic way.

-In paragraph 3, when mentioning biomarkers, it looks like the authors only mean "wet" biomarkers, not mentioning, for example, neurophysiological and neuroimaging biomarkers. Please, clarify.

-Please, provide references at the first mention of a given paper (for example, see line 316). Otherwise, the reader has to go through the whole paragraph to retrieve the possible reference a sentence is mentioning.

Author Response

We thank the reviewer for the dedicated time. We appreciated all the comments that we tried to reply properly and edit the manuscript. In detail: 
1) we shortened both paragraphs 2 and 3, as suggested; 
2) we added a summarizing table (in the manuscript as Table 2) to clarify the studies better; 
3) we specified that, following the review topic, we mentioned only the fluid biomarkers; 
4) when necessary, we added the references in the paragraph beginning. 

Round 2

Reviewer 1 Report

The article is revised well and can be accepted in its current format

Author Response

Many thanks. 

Reviewer 3 Report

I have no further comments. Please, check for minor spelling errors (e.g., svFDT and bvFDT in Figure 1, etc.)

Author Response

Many thanks. We updated the new figures.